# Safety Assessment of 3S, 3’S Astaxanthin Derived from Metabolically Engineered *K. marxianus*

**DOI:** 10.3390/antiox11112288

**Published:** 2022-11-18

**Authors:** Sabrina Yeo Samuel, Hui-Min David Wang, Meng-Yuan Huang, Yu-Shen Cheng, Juine-Ruey Chen, Wen-Hsiung Li, Jui-Jen Chang

**Affiliations:** 1Trade Wind Biotech Co., Ltd., Taipei 11574, Taiwan; 2Institute of Molecular Medicine, National Taiwan University, Taipei 10051, Taiwan; 3Institute of Biomedical Sciences, Academia Sinica, Taipei 11529, Taiwan; 4Graduate Institute of Biomedical Engineering, National Chung Hsing University, Taichung 40227, Taiwan; 5Graduate Institute of Medicine, College of Medicine, Kaohsiung Medical University, Kaohsiung 80708, Taiwan; 6Department of Medical Laboratory Science and Biotechnology, China Medical University, Taichung 40447, Taiwan; 7Department of Life Science, National Chung Hsing University, Taichung 40227, Taiwan; 8Department of Chemical and Materials Engineering, National Yunlin University of Science and Technology, Yunlin 64002, Taiwan; 9College of Future, National Yunlin University of Science and Technology, Yunlin 64002, Taiwan; 10RuenHuei Biopharmaceuticals Inc., Taipei 10050, Taiwan; 11Biodiversity Research Center, Academia Sinica, Taipei 11529, Taiwan; 12Department of Ecology and Evolution, University of Chicago, Chicago, IL 60637, USA; 13Graduate Institute of Integrated Medicine, China Medical University, Taichung 40447, Taiwan; 14Department of Medical Research, China Medical University Hospital, Taichung 40447, Taiwan

**Keywords:** antioxidant, probiotics, 3S, 3S′ astaxanthin, *Kluyveromyces marxianus*, food safety

## Abstract

Previous reviews have already explored the safety and bioavailability of astaxanthin, as well as its beneficial effects on human body. The great commercial potential in a variety of industries, such as the pharmaceutical and health supplement industries, has led to a skyrocketing demand for natural astaxanthin. In this study, we have successfully optimized the astaxanthin yield up to 12.8 mg/g DCW in a probiotic yeast and purity to 97%. We also verified that it is the desired free-form 3S, 3’S configurational stereoisomer by NMR and FITR that can significantly increase the bioavailability of astaxanthin. In addition, we have proven that our extracted astaxanthin crystals have higher antioxidant capabilities compared with natural esterified astaxanthin from *H. pluvialis*. We also screened for potential adverse effects of the pure astaxanthin crystals extracted from the engineered probiotic yeast by dosing SD rats with 6, 12, and 24 mg/kg/day of astaxanthin crystals via oral gavages for a 13-week period and have found no significant biological differences between the control and treatment groups in rats of both genders, further confirming the safety of astaxanthin crystals. This study demonstrates that developing metabolically engineered microorganisms provides a safe and feasible approach for the bio-based production of many beneficial compounds, including astaxanthin.

## 1. Introduction

Astaxanthin, or 3,3′-dihydroxy-β,β-carotene-4,4′-dione, belongs to the family of xanthophylls, which are the oxygenated derivatives of naturally occurring organic pigments called carotenoids [1]. Humans are unable to synthesize carotenoids on their own; hence, they depend on vegetables, fruits, and animal products to attain these valuable compounds. For astaxanthin, in particular, humans rely primarily on seafood, especially salmon. Other natural sources of astaxanthin include microalgae, yeasts, and different types of seafood, including krill and shrimps.

Recent research has indicated many beneficial effects of astaxanthin on human health, with astaxanthin being found to possess strong antioxidant [2], cardioprotective [3], neuroprotective [4], immunoprotective [5], and anti-cancer [6] effects. The great commercial potential in a variety of industries, such as the pharmaceutical and health supplement industries, has led to skyrocketing demands for natural astaxanthin. With the amount of research backing the beneficial functions of astaxanthin for the human body, there is now a developing market for nutraceutical astaxanthin that is expected to reach a market of several hundred million dollars in the near future.

Microalgae have long been utilized by humans as a vital source of nutrient-rich food, feed, and health-promoting substances. *Haematococcus pluvialis*, a commercial microalgae, is considered the richest source of natural astaxanthin, with 2–4% astaxanthin content in *Haematococcus* under stress conditions [7], which was why it was first used for the industrial-scale production of natural astaxanthin [8]. In addition, *H. pluvialis* biosynthesis predominantly produces the most valuable 3*S*, 3′*S* stereoisomers [9]. However, the industrial production of astaxanthin via microalgae is limited by the need for long cultivation periods, large surface areas, and significant capital needed for maintenance and infrastructure.

Natural astaxanthin can also be extracted from various micro-organisms, including the red yeast *Phaffia rhodozyma* (*Xanthophyllomyces dendrorhous*) [10]. However, despite their ability to reach high densities after brief propagation periods, these microbes possess low astaxanthin/carotenoid ratios, which decrease their economic feasibility, especially when compared with microalgal production. 

Astaxanthin can also be synthesized chemically, with synthetic astaxanthin chemically synthesized from petrochemicals in a highly involved, multistep process, where the molecules assume different forms before attaining the same chemical formula as natural astaxanthin [11]. However, despite sharing the same chemical formula, synthetic and natural astaxanthin differ substantially in three aspects: esterification, stereochemistry, and the presence of other naturally occurring carotenoids in natural astaxanthin. In addition, the bioavailability of synthetic astaxanthin is found to be much lower than that of natural astaxanthin [12]. Due to these differences, synthetic astaxanthin is currently not allowed for human use due to safety issues [13]. Instead, it is mostly utilized in fish feed to pigment the flesh of certain commercially farmed fish, primarily salmonids such as Atlantic salmon and trout.

To keep up with the increasing demand for natural astaxanthin in a sustainable way, efforts have been made to increase astaxanthin production in microorganisms through metabolic engineering due to the simplicity of genome engineering and short cell cycles. The construction of the metabolic pathway for astaxanthin production has been previously established in *Escherichia coli* K-12, a strain that has long been considered GRAS (Generally Recognized as Safe) and has a long history of safe commercial use in food applications. Genes encoding the enzymes needed for astaxanthin biosynthesis were cloned from the bacteria *Erwinia uredovora* and *Brevundimonas aurantiaca* into *E. coli* K-12 and had resulted in the generation of 5.8 mg/g DCW astaxanthin [14,15]. The astaxanthin extracted from *E. coli* K-12 has also been assessed via a 13-week study on subchronic toxicity in rats and accepted by both US Food and Drug Administration (USFDA) and Taiwan Food and Drug Administration (TFDA), further proving its safety for consumption [16], demonstrating that the biosynthesis of astaxanthin in metabolically engineered microorganisms can be a safe and feasible approach for the mass production of astaxanthin. 

Recent trends have used yeast cells such as *Pichia pastoris* or *Saccharomyces cerevisiae*, for the production of recombinant proteins in view of their increased yield and ability to synthesize functional eukaryotic proteins. Attempts at introducing the astaxanthin biosynthesis pathway into yeast have yielded considerable results, with 8.10 mg/g DCW being the highest astaxanthin yield reported to date in *S. cerevisiae* containing overexpressed genes [17]. 

With enhanced biomass production via the supplementation of excessive carbon sources [18], thermo-tolerant characteristics [19], appropriate glycosylation [20], and strong signal peptides [20], *K. marxianus* can potentially yield higher astaxanthin levels compared with *S. cerevisiae*. The construction of a more efficient astaxanthin biosynthesis pathway was attempted in *K. marxianus*, this time by expressing enzymes from algae, which yield the highest astaxanthin levels in nature. In addition, β-carotene ketolase (*bkt*) and hydroxylase (*hpchyb*) were also included to promote the conversion of beta-carotene into astaxanthin more efficiently. This resulted in the construction of an astaxanthin yield-improved yeast strain, S3−2, which can yield up to 3.125 mg/g DCW in a YPL medium and 5.701 mg/g DCW in a YPG medium, which is the highest astaxanthin content in an engineered host to date [21]. Biofunctional assessment of the astaxanthin extracted from the S3−2 *K. marxianus* strain, termed YEAST-astaxanthin, was performed to investigate the safety of the product, which was achieved by in vitro assays and two animal models. DPPH (2,2′-diphenyl-1-picrylhydrazyl radical) scavenging analysis, ferrous ion chelating ability, reducing power assessment, and mushroom tyrosinase inhibition evaluation indicated that YEAST-astaxanthin showed both antioxidative and tyrosine suppressive properties. The immersion of zebrafish larvae in YEAST-astaxanthin solution showed no significant toxic effects, while rats fed with YEAST-astaxanthin showed neither visible abnormalities nor substantial changes in body weight or blood biochemistry tests. In addition, rats fed with YEAST-astaxanthin also reported the inhibition of metastasis in lung melanoma cells and an increased survival rate [22]. 

In this study, we have successfully optimized the production of astaxanthin in *K. marxianus*, with a current yield of 12.9 mg/g DCW, and validated its purity and configuration. We also performed experimental tests to assess the biosafety of astaxanthin extracted from the metabolically engineered *K. marxianus*. A safety assessment was achieved by confirming the genome stability of astaxanthin-producing genes in the yeast and by performing a 13-week repeated dose oral toxicity study in rats. Our results indicate that the astaxanthin extracted from genetically engineered *K. marxianus* cells is indeed safe for consumption. 

## 2. Materials and Methods

### 2.1. HPLC 

The following mobile phases were employed with the Nomura Chemical Develosil C30-UG Column (Interlink Scientific Services, London, UK): buffer A, composed of methanol/MtBE/water (81:15:4 *v*/*v*/*v*), and buffer B, composed of methanol/MtBE/water (7:90:3 *v*/*v*/*v*). The flow rate of the mobile phase was 1 mL/ min., and the solvent gradient was as follows: from 0 to 45 min, 100% buffer A from 0 to 20 min, 100% buffer B from 20 to 21 min, and then 100% buffer A from 21 to 45 min. Samples were observed using a Jasco870-UV intelligent UV-VIS detector (JASCO International Co., Ltd., Tokyo, Japan). Using chromatography, commercially available reference chemicals, and a comparison of their spectra, all carotenoids were identified. Additionally, standards were used in combination with the extinction coefficients for quantification. 

### 2.2. UV Spectrometry 

The general pattern, maximum absorbance peaks, and wavelength range of the UV spectra of the isolated astaxanthin crystals from this study were compared. The extract was measured in the UV region (477.6 nm) with a PREMA PRO-739 spectrophotometer (Chuan Hua Precision, Taipei, Taiwan). 

### 2.3. Nuclear Magnetic Resonance Spectroscopy (NMR) 

A Jasco870-UV intelligent UV-VIS detector (JASCO International Co., Ltd., Tokyo, Japan) was previously used to identify extracted and purified astaxanthin. Ten milligrams of the pure astaxanthin was loaded into the Bruker Avance III HD 400 MHz NMR Spectrometer (Bruker Corporation, Massachusetts, USA) after being dissolved in 600 uL of DMSO-d6. The experiment’s parameters were (1) pulse program zg30; (2) number of scans, 64; and (3) relaxation time, 2 s. Finally, TopSpin software was used to examine the results. All NMR spectra were recorded on a Bruker Advance III 400 MHz. According to the residual proton resonances of the appropriate deuterated solvent, 1H NMR chemical shifts were reported relative to TMS. 

### 2.4. ABTS Assay 

The ABTS assay was conducted according to Arnao et al. (2001) [23], with a few adjustments. The stock solutions included both 2.6 mM potassium persulfate solution and 7.4 mM ABTSradical dot+ solution, which were combined in equal parts to create the working solution. It was then left to react for 12 h at room temperature in the dark before being diluted by combining 1mL of ABTSradical dot+ solution with 60 mL of methanol. This yielded an absorbance of 1.1 ± 0.02 units at 734 nm when checked by the spectrophotometer. For each assay, fresh ABTS radical dot+ solution was prepared. Fruit extracts (150 μL) were combined with 2850 L of the ABTS radical dot+ solution in the dark for 2 h. The spectrophotometer was then used to measure the absorbance at 734 nm. Between 25 and 600 M Trolox, the standard curve was linear. The units of measurement for the results were μM Trolox equivalents (TE)/g fresh mass. If the measured ABTS value was over the linear range of the standard curve, more dilution was required.

### 2.5. DPPH Assay 

The DPPH assay was performed according to the method of Brand-Williams et al. (1995) [24], with a few adjustments. An amount of 24 mg DPPH was combined with 100 mL of methanol to create the stock solution, which was then stored at −20 °C until needed. A stock solution of 10 mL was then mixed with 45 mL of methanol to obtain the working solution. This yielded an absorbance of 1.1 ± 0.02 units at 515 nm when checked by the spectrophotometer. Fruit extracts (150 μL) were combined with 2850 L of the ABTS radical dot+ solution in the dark for 24 h. The spectrophotometer was then used to measure the absorbance at 515 nm. Between 25 and 600 M Trolox, the standard curve was linear. The units of measurement for the results were μM Trolox equivalents (TE)/g fresh mass. If the measured ABTS value was over the linear range of the standard curve, more dilution was required.

### 2.6. Reducing Power Assay 

The testing samples were assessed for reducing property in accordance with a prior study [25]. By using a ferric 2,4,6-tripyridyl-S-triazine Fe(III)-TPTZ complex to produce a ferrous Fe(II)-TPTZ complex with a dark blue color using an adopted reductant, this technique measured the antioxidant’s decreasing ability. The positive control, 3-tert-butyl-4-hydroxyanisole (BHA) at 100 M, was utilized to detect the colorization at 700 nm. About 85 mL of phosphate-buffered saline (PBS) (67 mM, pH 6.8) and 2.5 L of 20% potassium ferricyanide (K_3_Fe(CN)_6_) were added to the test samples to slightly increase their volume. After 20 min of reaction time at 50 °C, 160 mL of 10% trichloroacetic acid was added to the reactants, and the mixture was centrifuged at 3000× *g* for 10 min. An amount of 25 uL FeCl_3_ (2%) was mixed with 75 mL of upper layer solution, and the optical density was measured at 700 nm. A higher optical absorption indicates a higher reductive property. 

### 2.7. Test Facility 

This study is designed according to the Safety Evaluation Methods for Health Food (2020), Ministry of Health and Welfare, Taiwan. In addition, the thirteen-week repeated dose oral toxicity study was conducted in compliance with the principles of Good Laboratory Practice for Non-clinical Laboratory Studies (FDA, 21 CFR, Part 58), the Good Laboratory Practice for Non-clinical Laboratory Studies (Ministry of Health and Welfare, ROC, 3rd ed., 2006), the OECD Principles on Good Laboratory Practice (TAF OECD GLP Compliance, No. 1, 1997), and the Regulations for Application of Health Food Permit (TFDA, 2016) outlined by the Ministry of Health and Welfare, Taiwan at the Preclinical Testing Center of Level Biotechnology Inc, New Taipei City, Taiwan. 

### 2.8. Test Substance 

The test substance, astaxanthin crystal (Asta-S Crystal, provided by Trade Wind Biotech Co. Ltd., Taipei, Taiwan), was extracted and purified from engineered *K. marxianus* biomass [22]. After fermentation, yeast cells were killed by heating at 90 °C and centrifuged to remove the spent medium. The concentrated astaxanthin-containing *K. marxianus* cells were then oven-dried before using ethyl acetate to extract the astaxanthin. Subsequently, residual biomass was then filtered out and crystallized by the addition of ethanol. Pure astaxanthin crystals as a food product were then collected using filter paper and analyzed to ensure quality control by performing assays for the astaxanthin content, solvent residue, heavy metals, DNA contamination, and microbiological safety. The astaxanthin crystal utilized in the toxicity study had a red crystalline appearance; was made of more than 98% total carotenoids according to a UV test; and included 97% (*w*/*w*) astaxanthin, as determined by HPLC. 

### 2.9. Animals and Treatment Regimen 

All experimental protocols were approved by the Institutional Animal Care and Use Committee (IACUC number: 210310) of Level Biotechnology Inc., New Taipei City, Taiwan. CD^®^ (Sprague Dawley) IGS rats were purchased from BioLASCO Ltd. (Yilan, Taiwan). The animals were housed in the AAALAC International accredited facility of Level Biotech. Inc. A total of 80 rats were randomly assigned to four study groups (vehicle control, low, mid, and high-dose groups; 10 animals/sex/group). The body weight variation of all animals used fell into an interval within ±20 percent of the mean weight. The basic design is presented in Appendix A.

According to the information provided by the sponsor, the human dose is 0.012 g per day (0.2 mg/kg for a 60 kg human). According to body weight conversion, the three dose levels selected in this study were approximately 30, 60, or 120-fold the human dose. The proposed human administration route is oral. Therefore, oral gavage was utilized in this study. The accuracies of the 1 or 3 mL syringes used in this study were the closest to 0.01 and 0.1 mL, respectively. The animals in all groups were dosed once daily for 91 consecutive days with the designated dose formulations (suspension) listed in the table. The first dosing day was denoted as Study Day 1 (Day 1). No animal replacement was performed. 

Throughout the experimental period, all animals were subjected to daily clinical monitoring following administration. The animals’ physical health, any relevant behavioral changes, and any overt poisoning symptoms were all noted. Observations were made on all animals’ overall health, skin, fur, eyes, and mucous membranes; the presence of fluids and excretions; and their autonomic activity (e.g., lacrimation, piloerection, and unusual respiratory pattern), among other things. The occurrence of chronic or tonic movements, stereotypies (such as excessive grooming or repetitive circling), and unusual behavior (such as self-mutilation or walking backward) were also noted, along with alterations in stride, posture, and responsiveness to handling. 

On the day of grouping and Day 91, an ophthalmologic examination was carried out on every animal. Using an ophthalmoscope, the cornea, conjunctiva, anterior chamber, iris, and lens were all inspected. Before necropsy, females had a single vaginal smear, and the estrus cycle was noted. After the predetermined dose time, hematology (including coagulation), serum chemistry, and urinalysis were carried out on all surviving animals. Blood was drawn through the abdominal aorta on the day of necropsy and divided into three tubes: one containing K2 EDTA for the analysis of the complete blood count, one containing sodium citrate for the analysis of the coagulation factors, and one containing no anticoagulant for the analysis of the serum chemistry. Prior to terminal sacrifice, urine samples were obtained by employing metabolism cages for 12–16 h, and the urine volume was recorded. The items analyzed can be found in Appendix A. 

The surviving animals on the day of sacrifice were anesthetized with the ketamine and xylazine mixture solution (80 mg/mL and 8 mg/mL, respectively), followed by blood collection, exsanguination, and necropsy. The gross necropsy includes examinations of the external surface of the body, all thoracic and abdominal cavities, intestines, and visceral organs. 

The sampled tissues and organs were kept after gross necropsy in 10% neutral buffered formalin or other suitable fixatives for a further histopathological investigation. Trimmed, embedded, sectioned, and H&E stained tissues from groups 1 (the vehicle control group) and 4 (the high-dose group) were chosen for microscopy analysis. These examinations were not extended to animals in group 2 (low-dose group) and group 3 (mid-dose group) because no treatment group-related findings were noted in the high-dose group. 

## 3. Results

### 3.1. Astaxanthin Compound Analysis

We first cultured the astaxanthin-producing strain *K. marxianus* S3−2 and then streaked it on a YPG medium plate to screen for a high-performing colony, named 6−13−a6. To ensure the stability of *K. marxianus* 6−13−a6, we first cultured it in YPG medium before sub-culturing it in 10 different medium mixtures for 10 generations and spreading them on YPG medium plates to obtain single colonies. These colonies were then screened to find the highest-performing colony, which was then used to conduct further experiments. This optimized strain was capable of producing up to 12.8 mg/g DCW of natural astaxanthin. In addition, we had also successfully achieved mass cultivation of the *K. marxianus* 6-13-a6 in commercial 10,000-L tanks, from which we were able to extract and purify astaxanthin crystals for further testing. In order to ensure the stability and purity of our extracted astaxanthin crystals, we conducted a series of experiments. Previously, we had already compared both freshly extracted astaxanthin crystals and extracted astaxanthin crystals stored for a year and had observed no noticeable differences in the color and luster of the crystals (data not shown). 

Astaxanthin exists in many different forms, ranging from stereoisomers, geometric isomers, and free to be esterified forms [2], and can be found in natural sources. The stereoisomers (3S, 3’S) and (3R 3’R) are the most abundant in nature. The (3S, 3’S) isomer is primarily found in *Haematococcus* algae, while the (3R, 3’R) isomer is mainly found in *Xanthophyllomyces dendrorhous*. However, a combination of all three isomers—(3S, 3’S), (3R, 3’S), and (3R, 3’R)—can be found in synthetic astaxanthin [8]. One of the most powerful known antioxidant properties is possessed by natural astaxanthin, which is regarded as a super antioxidant. Natural astaxanthin is 55× more potent than synthetic astaxanthin in trapping free radicals in our system [10,11]. Astaxanthin does, however, exist in *H. pluvialis* in three distinct forms that can be categorized as free (5%), monoesters (70%), and diesters (25%). Since esterified products are less ready to be digested and absorbed by the human body, this would affect the bioavailability of astaxanthin. In contrast, astaxanthin extracted from engineered probiotic yeast is mainly composed of free-form 3S, 3’S astaxanthin, which can significantly increase the bioavailability of astaxanthin [22]. Previous studies have also verified that the extracted astaxanthin is of the desired 3S, 3’S configurational stereoisomers by HPLC separation [22]. Therefore, to further ascertain the purity of the extracted astaxanthin crystals, we performed UV spectroscopy (477.6 nm) (Figure 1A) and High-Performance Liquid Chromatography (HPLC) (Figure 1B), where we were able to confirm its purity at 98.26% and 97.14%, respectively.

The ^1^H NMR (nuclear magnetic resonance) spectra and FTIR (Fourier transform infrared) spectra of the extracted astaxanthin are shown in Figure 2. As shown in Figure 2A,B, the ^1^H NMR spectrum analysis of astaxanthin indicates that there are multiple peaks in the chemical shift δ between 1.0 and 2.0, which are the characteristic absorption peaks of hydrogen in the -CH3, -CH2, and -CH configurations, while the chemical shifts around 2.0–3.0 and 5.05 are the characteristic absorption peaks of hydrogen in alkene’s -CH=CH2 configuration, and the chemical shifts at 6.0–7.0 are the characteristic absorption peaks of hydrogen on the benzene ring. Therefore, the test substance contains multiple structures such as -CH3, -CH2, -CH, -CH=CH2, and benzene rings. After analyzing the ^1^H NMR spectra of the astaxanthin standard (Sigma-Aldrich) and the extracted astaxanthin crystals, we are able to confirm that both materials possess the same characteristic absorption peak and, thus, have the same structural material composition.

In Figure 2C, a Fourier transform infrared spectroscopy (FTIR) analysis of the astaxanthin material structure shows peaks at 3031, 2918, and 2862 cm^−1^, which correspond to the symmetric and antisymmetric stretching vibration peaks of the -CH3, -CH2, and -CH configurations, while peaks at 1466 and 1386 cm^−1^ correspond to the stretching vibration peaks of the saturated C-H configuration, which indicates the presence of -CH3, -CH2, and -CH configurations in the sample. Peaks at 1733, 1648, and 1602 cm^−1^ are the stretching vibration peaks of C=O and alkene -CH=CH2 configurations, and the strong peaks generated at 974 and 953 cm^−1^ indicate that the samples contain multiple polyene structures. These analyses suggest that the extracted astaxanthin crystal has the same characteristic structures present in the astaxanthin standard, such as -CH3, -CH2, -CH, C=O, and polyene.

### 3.2. Antioxidant Activity Determinations

Many previous studies have already shown astaxanthin’s powerful antioxidant ability, which is essential for the absorption and neutralization of free radicals. In this study, we also verified the antioxidative capabilities of our extracted astaxanthin crystals using the ABTS assay, reducing power assay, and DPPH assay, which can detect the eradication of free radicals by observing color changes. The aforementioned assays are capable of distinguishing the different mechanisms used to eliminate reactive oxygen species (ROS): quenching of singlet and triplet oxygen or decomposing peroxides. Table 1 confirmed that every type of astaxanthin had dose-dependent competencies in the three experiments above, especially an ABTS scavenging ability, but barely had any differences in terms of reducing power.

The principle of ABTS assays in determining the antioxidant capacity is that ABTS (2,2’-azino-bis(3-ethylbenzothiazoline-6-sulfonic acid)) could be oxidized into green ABTS free radicals under the action of appropriate oxidants. Consequently, the generation of ABTS free radicals will be inhibited in the presence of antioxidants. Therefore, the change in the absorbance of ABTS can be measured and calculated as the total antioxidant capacity of the sample. Three types of astaxanthin showed the capabilities of eliminating ROS in ABTS assay. In general, in various concentrations, the extracted astaxanthin crystals were more potent than standard astaxanthin and algae astaxanthin. For example, the extracted astaxanthin crystals displayed a 60% scavenging rate at 1 mg/mL. Moreover, in low concentrations, the extracted astaxanthin crystals showed better capabilities, with a 32.68% oxidant removing speed while that for the standard astaxanthin was 25.1% and that for algae astaxanthin was 26.86%.

An established test for determining a substance’s ability to act as a hydrogen donor to eliminate the stable radical DPPH and to transform it into diphenyl-picrylhydrazine is the DPPH free radical scavenging system. When the element can absorb the free radicals, the assay color will change from deep blue to light yellow. Astaxanthin doses were directly correlated with the moderate antioxidant capacity displayed by all astaxanthin forms. For instance, the free radical emission capability was 50% in 1 mg/mL extracted astaxanthin crystals, which was higher than the exact dosage of standard astaxanthin (40.23%) and algae astaxanthin (43.68%). At 0.05 mg/mL of astaxanthin, the antioxidative ability was around 25% in three types of astaxanthin.

The reducing power assay is a common technique for determining the ability of materials in losing electrons by changing the color from yellow to green. The ability of the tested antioxidants is determined by the degree of color diversity. The presence of a suitable material allows for the reduction of the Fe^3+^/ferricyanide complex into a ferrous state. Table 1, which indicated no significant difference in the reducing power of the control group and various concentrations of astaxanthin, showed that no astaxanthin type exhibited any increased antioxidant reduction power. 

ROS promotes the secretion of a variety of inflammatory chemicals from cancer cells, which may increase the capacity of cancer cells to metastasize, boost the growth of vascular endothelium, and suppress the immune system. Due to ongoing oxidant production, ROS may also foster an immunosuppressive microenvironment for malignancies. Cancer cells can consequently continue to grow and degrade. Therefore, the biological role of antioxidants in reducing chronic inflammation and preventing cancer was implicated. 

The aforementioned tests demonstrated astaxanthin’s capacity to neutralize ROS, and it was superior in both the ABTS and DPPH assays. The reducing power likewise had a milder ability to reduce oxidation. Compared to the oxidative-eliminating abilities of all three types of astaxanthin, the extracted astaxanthin crystals performed much better than the standard and algae in both ABTS and DPPH in every concentration.

### 3.3. Thirteen Week Toxicity Study

In another study using astaxanthin produced by engineered *Escherichia coli* K-12, no significant biological differences between test and control groups was observed [15]. These results were consistent with prior toxicity studies using natural astaxanthin derived from bacteria or microalgae and even synthetic astaxanthin produced chemically. To further demonstrate the safety of astaxanthin derived from the engineered *K. marxianus* 6-13-a6, we conducted a 13-week repeated dose toxicity study to evaluate the potential toxicity likely to arise from repeated daily exposure of astaxanthin crystal in rats via oral gavage. The study’s findings provide information on human exposure safety.

#### 3.3.1. Mortality, Body Weight, Food Consumption, and Ophthalmological Examination

All of the animals survived the testing period and showed no toxicity-related symptoms (Table 2). When compared to the vehicle control group, neither male nor female rats in the treatment group showed statistically significant changes in body weight (Appendix A). The average food consumption for each group is shown in Appendix A, and for female rats, there was no statistically significant difference between the treatment groups and the vehicle control group. The mean food intake in male rats was considerably lower than the vehicle control over Days 50–57 at a dosage level of 12 mg/kg/day. Nonetheless, all values were within the historical control data range. Therefore, it was not considered a treatment group-related finding. No significant treatment group-related clinical signs were noted in male and female rats during the study period (Table 3). Several observed clinical signs such as chromodacryorrhea (1/10, high-dose males), hair loss (1/10 in the vehicle control females, 4/10 in the mid-dose females, and 1/10 in the high-dose females), and wound on the skin (1/10 in the low-dose females, 2/10 in the mid-dose females, and 1/10 in the high-dose females) were noted during the study period. Nonetheless, these clinical signs might be caused by social activities (i.e., fighting and grooming), and the incidence was not dose-dependent. Therefore, these findings were not considered a treatment group-related findings. In addition, ophthalmological examinations revealed no changes in all animals before grouping and terminal sacrifice (Appendix A). 

#### 3.3.2. Clinical Pathology 

No treatment-related toxicologically significant changes in any of the hematological parameters were seen compared with the control group (Table 4). In contrast to female rats, the value of PT (Prothrombin Time) was statistically higher at dose levels of 12 and 24 mg/kg/day in male rats, at 14.55 ± 1.57 s and 13.75 ± 2.06 s, respectively. In female rats, no statistical difference was observed when comparing test article-treated groups to the vehicle control group. In addition, APTT (activated Partial Thromboplastin Time) values were significantly higher at dose levels 6, 12, and 24 mg/kg/day compared with the vehicle control group, at 19.11 ± 1.24, 19.95 ± 1.19, and 19.30 ± 1.18, respectively. However, none of the values fell outside the range of the historical control data and were not regarded as treatment group-related observations.

The MCHC (mean corpuscular hemoglobin concentration) value was found to be statistically higher in male rats at the dose level of 24 mg/kg/day, with a value of 35.04 ± 0.37 g/dL, while EOS (eosinophil count) percentages were statistically lower in 12 and 24 mg/kg/day, with 0.26 ± 0.12% and 0.28 ± 0.14%, respectively. In contrast, in female rats, the MCHC value was statistically lower at dose levels of 12 and 24 mg/kg/day at 34.70 ± 0.35 g/dL and 34.75 ± 0.25 g/dL, respectively. Nonetheless, all the values were within the historical control data range. Since the changes were of a small magnitude, with no dose-dependency or correlated findings noted in HCT and HGB, it was not considered treatment group-related findings. 

Treatment with astaxanthin over a period of time showed no adverse effects on the clinical chemistry parameters (Table 4). Nevertheless, statistically significant differences were observed between vehicle control groups and treatment groups in male rats: the values of CHOL (cholesterol) and TG (triglycerides) at dose levels of 6 and 24 mg/kg/day were statistically lower than those in the vehicle control group at 53.31 ± 11.14 mg/dL and 21.51 ± 6.26 mg/dL, respectively. However, all values were within the historical control data range and were not considered treatment group-related findings. No statistical difference was noted when comparing the treatment group to the vehicle control group in female rats. 

Both male and female rats’ urinalysis results showed no treatment-related toxicologically significant changes (Table 4). 

#### 3.3.3. Relative Organ Weights

In both male and female rats, no treatment group-related finding was noted in relative organ weight (organ-to-brain weight) data (Appendix A). 

#### 3.3.4. Gross Necropsy Findings 

The gross necropsy findings are summarized in Appendix A. In males, gross findings such as discoloration in the thymus (1/10) and adrenal glands (1/10) and abnormal shape in the brain (1/10) were noted in the vehicle control on Day 92. In females, discoloration in the mandibular lymph node (1/10) and nodule in the liver (1/10) were indicated in the vehicle control on Day 92. Nonetheless, these findings were not treatment-related due to the lack of dose dependency.

#### 3.3.5. Histopathology Evaluation

No signs of necropsy or other histopathologic findings (Appendix A) related to astaxanthin exposure were found in the control or treatment groups. While several findings, including congestion of the thymus and mandibular lymph node, mononuclear cell infiltration and fatty change in the liver, tubular epithelial cell degeneration, interstitial mononuclear cell infiltration, and mineralization of the kidneys were noted, these findings were incidental and were not considered treatment-related findings.

## 4. Discussion

With the rapid development of revolutionary technology, synthetic biology will soon play an essential role in future industries; whether it is energy, medicine, chemistry, food, biodegradable consumable products, or even medical testing, it will soon be deeply involved in the industry supply chain. However, many challenges remain before genetically modified organisms and its products can become mainstream on the market. One of the biggest challenges is the acceptance of anything genetically modified by the general public. Despite numerous studies proving that GM products are similar if not even better than the conventional non-GM products, many people are still wary. 

Thus, we aim to alleviate these worries by the public. We used *K. marxianus*, an emerging non-conventional food-grade yeast, as our chassis. Additionally, since this particular yeast has aerobic-respiring characteristics [26], which will not produce any ethanol byproducts, our product can also be marketed as halal. To alleviate any concerns about genetic modification of the yeast, we also extracted and purified the desired astaxanthin. In our study, we have validated the purity and configuration of our extracted astaxanthin crystals and found that it still maintains the highly sought after high antioxidative function. In addition, we also assessed the safety of consuming our extracted astaxanthin crystals. Minor changes can be seen in mean food consumption, clinical observations, PT values, APTT values, MCHC values, EOS percentages, cholesterol values, triglycerides values, and gross necropsy observations. Nonetheless, these findings were not treatment-related in view of the lack of dose dependency. Our results were also comparable with several other coordinated studies on the subchronic toxicity of astaxanthin conducted by Katsumata et al. (2014) [27] and Lin et al. (2017) [15], which employed astaxanthin extracted from native bacteria *P. carotinifaciens* and engineered *E. coli,* respectively. In addition, subchronic toxicity of chemically synthesized astaxanthin and *H. pluvialis* biomass, which is rich in astaxanthin esters, were assessed by Vega et al. (2015) [28] and Stewart et al. (2008) [29], respectively, which revealed no significant biological changes.

Previous reviews have already explored the safety and bioavailability of astaxanthin, as well as the beneficial effects it has on human body [30,31]. Our findings have further confirmed the safety of astaxanthin produced by probiotic yeast. 

## 5. Conclusions

This study provided an example that developing metabolically engineered microorganisms provides a safe and feasible approach for the bio-based mass production of beneficial compounds. Optimizing and engineering the *K. marxianus* strain for astaxanthin production has indeed allowed for record breaking astaxanthin production, which trumps even the natural astaxanthin producer *H. pluvialis* while maintaining the desired configuration and function. Screening of the potential adverse effects of astaxanthin crystals extracted from engineered *K. marxianus* 6-13-a6 by dosing SD rats with 6, 12, and 24 mg/kg/day of astaxanthin crystals by oral gavages for a 13-week period has indicated no significant biological changes between control and treatment groups in rats of both genders, further confirming the safety of the astaxanthin crystals. 

## Figures and Tables

**Figure 1 antioxidants-11-02288-f001:**
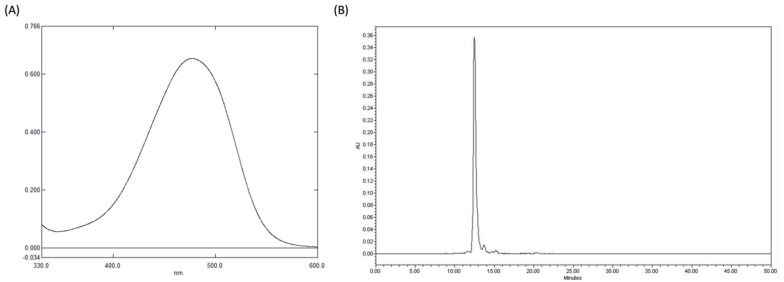
Validation of astaxanthin purity extracted and purified from the strain *K. marxianus* 6−13−a6. (**A**) Evaluation of astaxanthin purity by UV spectrometer. (**B**) Evaluation of astaxanthin purity by HPLC.

**Figure 2 antioxidants-11-02288-f002:**
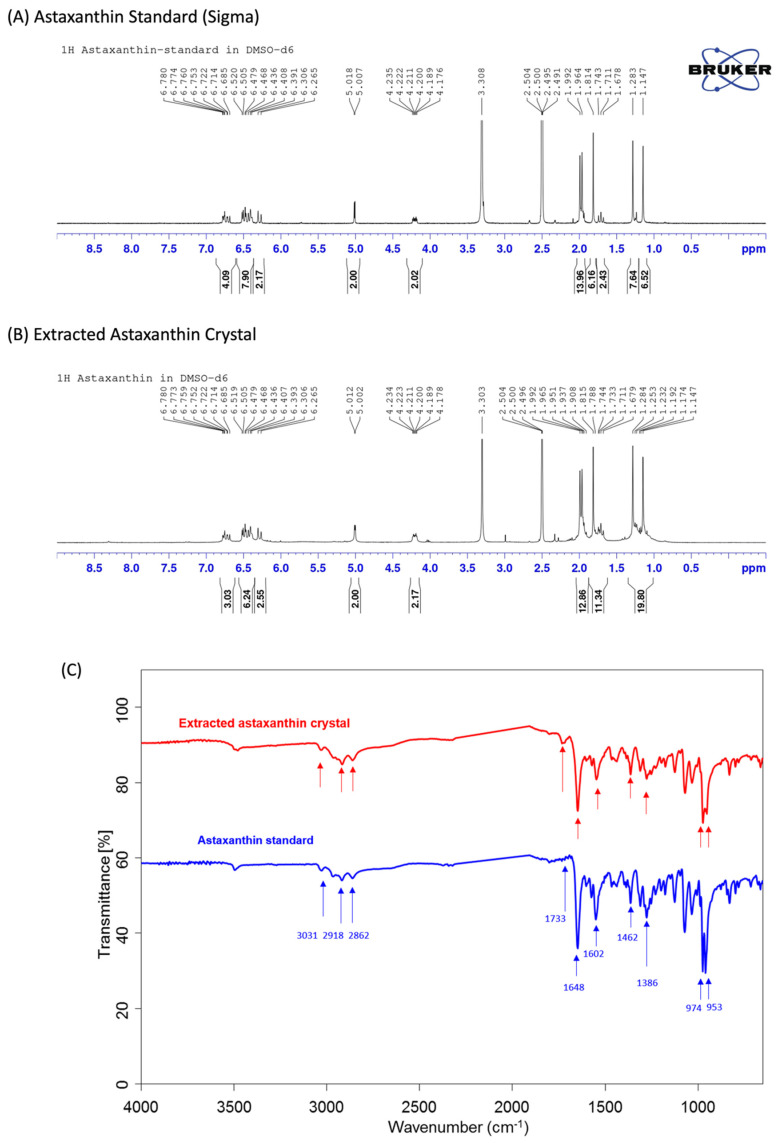
Verification of astaxanthin chemical composition extracted from *K. marxianus* 6−13−a6 by NMR (**A**,**B**) and FTIR (**C**).

**Table 1 antioxidants-11-02288-t001:** ABTS, reducing power, and DPPH were used to test the ROS reduce abilities in various astaxanthin. Three types of astaxanthin: Sigma-Aldrich Co., Burlington, MA, USA; extracted astaxanthin; and algae.

Concentration (µg/mL)		Antioxidant Capacity of Astaxanthin		
		ABTS (%)	DPPH (%)	Reducing Power (Absorbance at 700 nm)
Positive control ^a,b,c^		93.4 ± 0.003	88.14 ± 0.008	0.66 ± 0.012
Standard astaxanthin (Sigma-Aldrich)	0.05	25.10 ± 020	24.20 ± 0.001	0.107 ± 0.001
	0.1	39.54 ± 0.006	32.21 ± 0.005	0.109 ± 0.002
	1	55.10 ± 0.0033	40.23 ± 0.003	0.122 ± 0.005
Extracted astaxanthin crystals	0.05	32.68 ± 0.034	30.22 ± 0.035	0.109 ± 0.002
	0.1	47.42 ± 0.028	42.30 ± 0.006	0.112 ± 0.006
	1	60.31 ± 0.06	60.23 ± 0.012	0.223 ± 0.005
Algae astaxanthin	0.05	26.86 ± 0.006	25.04 ± 0.002	0.109 ± 0.003
	0.1	40.21 ± 0.017	38.11 ± 0.014	0.108 ± 0.002
	1	51.08 ± 0.023	43.68 ± 0.002	0.133 ± 0.001

^a^. Vitamin C was used as a positive control on ABTS at 2 mM. ^b^. Vitamin C was used as a positive control on the DPPH assay at 10 mM. ^c^. BHA was used as a positive control on reducing power at 10 mM.

**Table 2 antioxidants-11-02288-t002:** Mortality of rats receiving astaxanthin over a 13-week period.

Gender	Dose (mg/kg/day)	Mortality (N/N) ^1^
Male	0	0/10
	6	0/10
	12	0/10
	24	0/10
Female	0	0/10
	6	0/10
	12	0/10
	24	0/10

^1^ N/N: Number of animals found dead/total number of animals.

**Table 3 antioxidants-11-02288-t003:** Clinical observations of rats receiving astaxanthin over a 13-week period.

Gender	Clinical Signs	Incidence (n’/n’) ^1^			
		Vehicle Control 0 mg/kg/day	Low-Dose 6 mg/kg/day	Mid-Dose 12 mg/kg/day	High-Dose 24 mg/kg/day
		*n* = 10	*n* = 10	*n* = 10	*n* = 10
Male	Chromodacryorrhea	0/10	0/10	0/10	1/10
Female	Hair loss	1/10	0/10	4/10	1/10
	Wound, skin	0/10	1/10	2/10	1/10

^1^ Total number of animals with observable sign/total number of animals examined.

**Table 4 antioxidants-11-02288-t004:** (A) Coagulation analysis of rats receiving astaxanthin over a 13-week period. (B) Hematology values of rats receiving astaxanthin over a 13-week period. (C) Serum chemistry of rats receiving astaxanthin over a 13-week period. (D) Urinalysis of rats receiving astaxanthin over a 13-week period.

**A**								
**Parameters**	**Coagulation (Mean ± SD)**							
**Group**	**Vehicle Control** **0 mg/kg/day**	**Low-Dose** **6 mg/kg/day**	**Mid-Dose** **12 mg/kg/day**	**High-Dose** **24 mg/kg/day**	**Vehicle Control** **0 mg/kg/day**	**Low-Dose** **6 mg/kg/day**	**Mid-Dose** **12 mg/kg/day**	**High-Dose** **24 mg/kg/day**
	**Male**				**Female**			
**Number of Animals**	**10**	**10**	**10**	**10**	**10**	**10**	**10**	**10**
PT (sec)	11.95 ± 1.30	13.27 ± 1.26	14.55 ± 1.57 *	13.75 ± 2.06 *	9.40 ± 0.51	9.11 ± 0.25	9.28 ± 0.27	9.25 ± 0.15
APTT (sec)	17.33 ± 2.12	19.11 ± 1.24 *	19.95 ± 1.19 *	19.30 ± 1.18 *	15.58 ± 1.00	15.58 ± 1.06	16.23 ± 0.63	16.14 ± 1.12
FIB (mg/dL)	250.33 ± 20.57	238.66 ± 24.42	237.01 ± 17.83	230.77 ± 14.00	169.19 ± 26.81	179.75 ± 9.73	179.30 ± 11.35	184.23 ± 6.15
*: *p* ≤ 0.05 (compared to vehicle control group)								
**B**								
**Parameters**	**Hematology (Mean ± SD)**							
**Group**	**Vehicle Control** **0 mg/kg/day**	**Low-Dose** **6 mg/kg/day**	**Mid-Dose** **12 mg/kg/day**	**High-Dose** **24 mg/kg/day**	**Vehicle Control** **0 mg/kg/day**	**Low-Dose** **6 mg/kg/day**	**Mid-Dose** **12 mg/kg/day**	**High-Dose** **24 mg/kg/day**
	**Male**				**Female**			
**Number of Animals**	**10**	**10**	**10**	**10**	**10**	**10**	**10**	**10**
WBC (10^3^/mL)	6.795 ± 1.627	6.146 ± 1.513	6.666 ± 1.470	5.904 ± 1.592	4.692 ± 1.857	4.364 ± 1.767	5.243 ± 1.342	4.668 ± 0.543
RBC (10^6^/mL)	9.002 ± 0.389	9.360 ± 0.440	9.279 ± 0.326	9.039 ± 0.364	7.985 ± 0.340	7.845 ± 0.135	8.189 ± 0.315	7.899 ± 0.256
HGB (g/dL)	15.77 ± 0.70	16.36 ± 0.61	16.40 ± 0.28	16.16 ± 0.70	14.64 ± 0.53	14.42 ± 0.34	14.89 ± 0.42	14.63 ± 0.50
HCT (%)	45.59 ± 1.91	46.94 ± 1.56	47.31 ± 1.10	46.12 ± 1.92	41.59 ± 1.61	41.31 ± 0.70	42.92 ± 1.27	42.10 ± 1.25
MCV (fL)	50.68 ± 1.43	50.20 ± 1.40	51.04 ± 1.66	51.04 ± 1.46	52.10 ± 1.27	52.67 ± 1.21	52.45 ± 1.56	53.33 ± 0.97
MCH (pg)	17.53 ± 0.53	17.48 ± 0.61	17.68 ± 0.53	17.88 ± 0.52	18.33 ± 0.46	18.38 ± 0.58	18.18 ± 0.50	18.53 ± 0.35
MCHC (g/dL)	34.59 ± 0.24	34.85 ± 0.39	34.66 ± 0.30	35.04 ± 0.37 *	35.20 ± 0.32	34.90 ± 0.44	34.70 ± 0.35*	34.75 ± 0.25 *
PLT (10^3^/mL)	1065.3 ± 81.9	1075.9 ± 166.1	1057.7 ± 106.9	1084.3 ± 80.6	1015.7 ± 138.0	940.4 ± 228.5	1027.8 ± 79.9	962.5 ± 110.6
NEU (%)	26.76 ± 11.05	27.61 ± 5.12	24.08 ± 9.26	28.85 ± 9.82	15.85 ± 5.84	12.41 ± 3.26	15.49 ± 5.00	13.04 ± 5.28
LYM (%)	65.57 ± 10.33	65.52 ± 5.25	68.80 ± 9.51	63.74 ± 9.73	78.13 ± 6.61	81.97 ± 4.62	79.32 ± 5.72	81.51 ± 6.03
MON (%)	6.81 ± 1.10	6.24 ± 0.65	6.64 ± 1.11	6.94 ± 1.16	5.42 ± 1.91	5.21 ± 1.37	4.92 ± 1.68	5.03 ± 1.09
EOS (%)	0.61 ± 0.47	0.40 ± 0.24	0.26 ± 0.12 *	0.28 ± 0.14 *	0.40 ± 0.33	0.34 ± 0.26	0.16 ± 0.14	0.28 ± 0.21
BAS (%)	0.25 ± 0.08	0.23 ± 0.11	0.22 ± 0.12	0.19 ± 0.09	0.20 ± 0.13	0.07 ± 0.19	0.11 ± 0.11	0.14 ± 0.10
RET (%)	2.214 ± 0.28	2.143 ± 0.382	2.206 ± 0.173	2.162 ± 0.309	2.088 ± 0.472	2.261 ± 0.413	1.982 ± 0.338	2.321 ± 0.362
*: *p* ≤ 0.05 (compared to vehicle control group)								
**C**								
**Parameters**	**Serum Chemistry (Mean ± SD)**							
**Group**	**Vehicle Control** **0 mg/kg/day**	**Low-Dose** **6 mg/kg/day**	**Mid-Dose** **12 mg/kg/day**	**High-Dose** **24 mg/kg/day**	**Vehicle Control** **0 mg/kg/day**	**Low-Dose** **6 mg/kg/day**	**Mid-Dose** **12 mg/kg/day**	**High-Dose** **24 mg/kg/day**
	**Male**				**Female**			
**Number of Animals**	**10**	**10**	**10**	**10**	**10**	**10**	**10**	**10**
AST (U/L)	99.25 ± 18.36	103.85 ± 22.32	93.14 ± 25.58	88.91 ± 12.18	77.66 ± 11.13	81.93 ± 16.99	79.52 ± 13.35	73.12 ± 12.26
ALT (U/L)	28.82 ± 3.59	31.71 ± 4.92	27.02 ± 6.61	28.38 ± 3.02	25.57 ± 3.51	25.57 ± 8.28	22.64 ± 3.33	22.01 ± 5.33
GLU (mg/dL)	222.80 ± 51.18	201.05 ± 47.60	200.08 ± 20.95	193.29 ± 40.03	171.20 ± 29.06	188.49 ± 43.42	171.36 ± 16.88	180.18 ± 32.94
TP (g/dL)	5.92 ± 0.43	5.93 ± 0.40	5.84 ± 0.24	5.85 ± 0.33	6.22 ± 0.34	6.10 ± 0.36	6.38 ± 0.41	6.15 ± 0.32
ALB (g/dL)	3.94 ± 0.27	3.97 ± 0.23	3.92 ± 0.20	3.82 ± 0.21	4.55 ± 0.28	4.48 ± 0.34	4.52 ± 0.37	4.42 ± 0.24
TBIL (mg/dL)	0.063 ± 0.015 (n = 9 ^a^)	0.063 ± 0.019 (n = 9 ^a^)	0.063 ± 0.013 (n = 9 ^a^)	0.062 ± 0.017 (n = 9 ^a^)	0.071 ± 0.019	0.082 ± 0.020 (n = 9 ^a^)	0.063 ± 0.013 (n = 8 ^a^)	0.071 ± 0.014
BUN (mg/dL)	13.90 ± 1.97	14.28 ± 1.52	13.43 ± 1.86	13.60 ± 0.76	14.31 ± 2.42	14.77 ± 2.26	13.48 ± 1.20	13.38 ± 1.58
CREA (mg/dL)	0.34 ± 0.07	0.35 ± 0.07	0.33 ± 0.05	0.33 ± 0.05	0.41 ± 0.06	0.44 ± 0.07	0.40 ± 0.07	0.40 ± 0.05
GGT (U/L)	<2	<2	<2	<2	<2	<2	<2	<2
ALP (U/L)	221.78 ± 29.19	226.35 ± 32.10	246.24 ± 38.71	247.44 ± 56.76	122.43 ± 28.33	114.42 ± 33.98	131.72 ± 52.40	125.49 ± 46.07
CHOL (mg/dL)	68.44 ± 18.01	53.91 ± 10.23 *	55.24 ± 7.85	53.31 ± 11.14 *	52.22 ± 10.91	62.32 ± 18.28	61.47 ± 13.94	53.50 ± 14.25
TG (mg/dL)	37.60 ± 19.36	22.68 ± 6.82 *	25.64 ± 9.37	21.51 ± 6.26 *	18.59 ± 6.85	20.89 ± 6.94	15.05 ± 4.51	18.59 ± 8.67
Ca (mg/dL)	9.93 ± 0.32	9.73 ± 0.32	9.84 ± 0.32	9.71 ± 0.29	10.07 ± 0.26	9.99 ± 0.35	10.02 ± 0.21	10.03 ± 0.24
P (mg/dL)	6.50 ± 0.58	6.63 ± 0.64	6.59 ± 0.28	6.32 ± 0.45	5.41 ± 0.79	5.56 ± 0.57	5.74 ± 0.84	5.92 ± 0.69
CK (U/L)	447.55 ± 326.04	433.81 ± 305.08	394.79 ± 340.79	367.31 ± 185.55	270.22 ± 150.71	285.67 ± 133.62	276.82 ± 182.04	249.60 ± 97.66
AMY (U/L)	1557.9 ± 240.9	1395.3 ± 269.0	1368.0 ± 148.0	1338.0 ± 182.3	872.4 ± 159.7	943.9 ± 210.7	991.5 ± 149.2	948.4 ± 309.0
Na (mmol/L)	143.41 ± 2.04	143.59 ± 1.28	143.76 ± 1.47	143.50 ± 0.96	142.17 ± 0.56	142.25 ± 1.16	142.61 ± 1.44	142.55 ± 1.49
K (mmol/L)	4.480 ± 0.432	4.390 ± 0.396	4.380 ± 0.204	4.350 ± 0.310	4.330 ± 0.340	4.410 ± 0.307	4.490 ± 0.446	4.280 ± 0.193
Cl (mmol/L)	105.50 ± 1.34	105.75 ± 1.49	105.62 ± 1.31	106.90 ± 1.93	105.79 ± 1.36	106.58 ± 1.26	106.31 ± 0.92	107.04 ± 1.72
LDH (U/L)	737.2 ± 459.1	653.7 ± 375.6	737.4 ± 683.5	673.0 ± 389.8	528.5 ± 315.9	513.6 ± 241.8	611.6 ± 440.1	502.8 ± 228.2
TBA (umol/L)	6.9 ± 2.6	10.7 ± 7.7	9.1 ± 3.6	12.2 ± 10.5	9.4 ± 3.2	18.0 ± 22.4	11.0 ± 3.3	8.9 ± 1.7
^a^ The values of TBIL in other animals were lower than the detection limit. *: *p* ≤ 0.05 (compared to vehicle control group)								
**D**								
**Gender**	**Parameters**	**Urine Quantitative Analysis (Mean ± SD)**						
	**Group**	**Vehicle Control** **0 mg/kg/day**		**Low-Dose** **6 mg/kg/day**	**Mid-Dose** **12 mg/kg/day**		**High-Dose** **24 mg/kg/day**	
	**Number of Animals**	**10**		**10**	**10**		**10**	
Male	Volume (mL)	17.70 ± 7.65		17.50 ± 5.85	16.60 ± 7.06		16.20 ± 9.58	
	SG	1.0178 ± 0.0036 (n = 9 ^a^)		1.0180 ± 0.0035	1.0172 ± 0.0036 (n = 9 ^a^)		1.0190 ± 0.0046	
	pH	7.35 ± 0.63		7.00 ± 0.24	7.30 ± 0.35		7.20 ± 0.26	
	UURO (EU/dL)	0.28 ± 0.25		0.20 ± 0.00	0.20 ± 0.00		0.20 ± 0.00	
Female	Volume (mL)	12.00 ± 5.29		8.30 ± 4.11	9.20 ± 3.46		12.00 ± 6.99	
	SG	1.0170 ± 0.0042		1.0189 ± 0.0042 (n = 9 ^a^)	1.0183 ± 0.0035 (n = 9 ^a^)		1.0185 ± 0.0047	
	pH	6.65 ± 0.47		6.95 ± 0.55	6.75 ± 0.59		7.00 ± 0.24	
	UURO (EU/dL)	0.20 ± 0.00		0.20 ± 0.00	0.20 ± 0.00		0.20 ± 0.00	
^a^ The values of specific gravity from other animals were higher than the detection limit.								

## Data Availability

All data generated or analyzed during this study are included in the published article (and its online Appendix A).

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
