# Peer review of "Safety Assessment of 3S, 3’S Astaxanthin Derived from Metabolically Engineered K. marxianus"

_antioxidants, 2022, doi:10.3390/antiox11112288_

Round 1
Reviewer 1 Report (Previous Reviewer 1)
This is a revised version of a manuscript I reviewed several days ago. The authors have addressed my initial comments on microbial sections.
Author Response
We appreciate the time and effort that you have dedicated to providing feedback on our manuscript and are grateful for the insightful comments on and valuable improvements to our paper.
Reviewer 2 Report (Previous Reviewer 2)
1. There is no need to list experimental methods in the abstract, and more important results, numbers details and significance of the research should be added.
2. massive English editing and modification are needed for considering further process.
3. Please avoid using vertical rules and shading in Table. Additionally, some forms exist as pictures, please edit according to the correct form of the journal and kindly ensure it is in the journal’s style.
Author Response
We appreciate the time and effort that you have dedicated to providing feedback on our manuscript and are grateful for the insightful comments on and valuable improvements to our paper. We hope the manuscript, after careful revisions, meets your high standards. We welcome further constructive comments if any. Below we provide the point-by-point responses.
1. There is no need to list experimental methods in the abstract, and more important results, numbers details and significance of the research should be added.
Response: Thank you very much for the reminder. We have made revisions accordingly.
2. Massive English editing and modification are needed for considering further process.
Response: Thank you very much for the reminder. We have made revisions accordingly. In addition, we had a native English speaker review our manuscript before the submission.
3. Please avoid using vertical rules and shading in Table. Additionally, some forms exist as pictures, please edit according to the correct form of the journal and kindly ensure it is in the journal’s style.
Response: Thank you very much for the reminder. We have ensured no vertical rules and shading are applicable in our tables.
Round 2
Reviewer 2 Report (Previous Reviewer 2)
Manuscript entitled “Safety assessment of 3S, 3S' Astaxanthin derived from metabolically engineered K. marxianus” has adequately addressed the previous concerns and the re-submission has been greatly improved, it’s comprehensive to convince me for accepting the manuscript.
This manuscript is a resubmission of an earlier submission. The following is a list of the peer review reports and author responses from that submission.
Round 1
Reviewer 1 Report
The manuscript antioxidants-1927595 mainly talks about the safety assessment of 3S, 3S' Astaxanthin derived from metabolically engineered K. marxianus, including the antioxidant activity and toxicity. The results suggest that 3S, 3S' Astaxanthin derived from metabolically engineered K. marxianus exhibits high antioxidant activity and safety. I am not good at the medicine field and food safety field. From a scientist of microbiology/biotechnology, I think there are several improper issues in this manuscript, and should be revised.
1. Line 61-62, Natural astaxanthin can also be extracted from various bacteria, including the red yeast Phaffia rhodozyma (Xanthophyllomyces Dendrorhous). The information is incorrect. Phaffia rhodozyma is not a bacteria.
2. Line 100, unlike S. cerevisiae, the K. marxianus is considered GRAS. The information is incorrect. S. cerevisiae is also GRAS.
3. Line 104-107, S3-2 can yield up to 3.125 mg/g DCW in a YPL medium and 5.701 mg/g DCW in a YPG medium, which is the highest astaxanthin content in an engineered host to date. Obviously, this information is incorrect. As mentioned in the manuscript, engineered S. cerevisiae contributes to at least 8.10 mg/g, and engineered E. coli contributed to 5.8 mg/g DCW astaxanthin.
4. The references in the “Materials and Methods” and “Results” sections are missed from the “References” section.
5. Line 215-216, A total of 248 80 rats were randomly assigned to four study groups (vehicle control, low, mid, and high-dose groups, 10 animals/sex/group). The rats number for this study is 24880?
6. Line 455-456, This study is the first to use probiotic yeast as a cell factory for the high-purity bioproduction of natural astaxanthin. The information is not suitable for this manuscript. First, astaxanthin production from a cell factory is not the content of this study, instead, this study mainly investigate the safety assessment of astaxanthin; Second, K. marxianus is not the first yeast cell factory for astaxanthin production.
7. Line 482-493. This paragraph mainly talks about the advantages of producing astaxanthin from microorganisms. Nonetheless, this is not the topic of this study. The authors should pay more attention on the safety assessment of astaxanthin.
Reviewer 2 Report
1. There is no need to list experimental methods in the abstract, and more important results, numbers details and significance of the research should be added. The results of the paper, such as the chemical composition and antioxidant capacity of astaxanthin, are not described in detail in the abstract.
2. Please explain the novelty of this manuscript, has anyone studied the safety assessment of astaxanthin derived from metabolically engineered K. marxianus before? many concerns need to be further complemented in introduction.
3. Research on the Clinical Pathology of the astaxanthin could be further studied in this study.
4. Please avoid using vertical rules and shading in all tables, please edit according to the correct form of the journal and kindly ensure it is in the journal’s style.
5. A lot of description and statement are not easy to understand; thus massive English editing and modification are needed for considering further process.